# Influence of Chronic Ocular Hypertension on Emmetropia: Refractive, Structural and Functional Study in Two Rat Models

**DOI:** 10.3390/jcm10163697

**Published:** 2021-08-20

**Authors:** Silvia Mendez-Martinez, Teresa Martínez-Rincón, Manuel Subias, Luis E. Pablo, David García-Herranz, Julian García Feijoo, Irene Bravo-Osuna, Rocío Herrero-Vanrell, Elena Garcia-Martin, María J. Rodrigo

**Affiliations:** 1Department of Ophthalmology, Miguel Servet University Hospital, 50009 Zaragoza, Spain; teresamrincon@gmail.com (T.M.-R.); manusubias@gmail.com (M.S.); lpablo@unizar.es (L.E.P.); egmvivax@yahoo.com (E.G.-M.); mariajesusrodrigo@hotmail.es (M.J.R.); 2Miguel Servet Ophthalmology Research Group (GIMSO), Aragon Health Research Institute (IIS Aragon), 50009 Zaragoza, Spain; 3National Ocular Pathology Network (OFTARED), Carlos III Health Institute, 28040 Madrid, Spain; jgarciafeijoo@hotmail.com (J.G.F.); ibravo@ucm.es (I.B.-O.); rociohv@farm.ucm.es (R.H.-V.); 4Innovation, Therapy and Pharmaceutical Development in Ophthalmology (InnOftal) Research Group, UCM 920415 Department of Pharmaceutics and Food Technology, Faculty of Pharmacy, Complutense University of Madrid, 28040 Madrid, Spain; davgar07@ucm.es; 5Health Research Institute, San Carlos Clinical Hospital (IdISSC), 28040 Madrid, Spain; 6University Institute for Industrial Pharmacy (IUFI), School of Pharmacy, Complutense University of Madrid, 28040 Madrid, Spain; 7Department of Ophthalmology, San Carlos Clinical Hospital (IdISSC), Complutense University of Madrid, 28040 Madrid, Spain

**Keywords:** glaucoma, intraocular pressure, refractive error, neuroretina, myopia

## Abstract

Chronic ocular hypertension (OHT) influences on refraction in youth and causes glaucoma in adulthood. However, the origin of the responsible mechanism is unclear. This study analyzes the effect of mild-moderate chronic OHT on refraction and neuroretina (structure and function) in young-adult Long-Evans rats using optical coherence tomography and electroretinography over 24 weeks. Data from 260 eyes were retrospectively analyzed in two cohorts: an ocular normotension (ONT) cohort (<20 mmHg) and an OHT cohort (>20 mmHg), in which OHT was induced either by sclerosing the episcleral veins (ES group) or by injecting microspheres into the anterior chamber. A trend toward emmetropia was found in both cohorts over time, though it was more pronounced in the OHT cohort (*p* < 0.001), especially in the ES group (*p* = 0.001) and males. IOP and refraction were negatively correlated at week 24 (*p* = 0.010). The OHT cohort showed early thickening in outer retinal sectors (*p* < 0.050) and the retinal nerve fiber layer, which later thinned. Electroretinography demonstrated early supranormal amplitudes and faster latencies that later declined. Chronic OHT accelerates emmetropia in Long–Evans rat eyes towards slowly progressive myopia, with an initial increase in structure and function that reversed over time.

## 1. Introduction

An increase in intraocular pressure (IOP) is the main factor for developing primary open-angle glaucoma (POAG), which leads to chronic progressive optic neuropathy and is a leading cause of irreversible blindness worldwide [1]. An association between POAG and high myopia was found, as highly myopic patients experience increased risk and earlier onset [2] of this sight-threatening disease [3,4,5,6]. The prevalence of myopia and high myopia (spherical equivalent less than −6.00 D) in the global population is currently 28.3%, and this is expected to increase by 2050 [7]. Highly myopic eyes suffer from structural changes such as peripheral retinal atrophic areas, tilted nerve appearance, temporal crescent of the optic disc [8], vitreoretinal traction, retinoschisis, lamellar or complete macular hole, retinal pigment epithelium (RPE) alterations and myopic choroidal neovascularization, lacquer cracks and chorioretinal atrophy, decreased choroidal thickness, scleral thinning with irregular curvature and staphyloma [9], and decreased retinal nerve fiber layer (RNFL) thickness, among others [10]. All these structural differences in myopic eyes are probably related to biomechanical stretching resulting from the imbalance between IOP and the elastic properties of the sclera [11,12,13,14]. Several animal and human studies have claimed the baropathic nature of axial myopia [15]. Indeed, several topical ocular hypotensive drugs were used in animal [16,17] and in human [18] studies to demonstrate the influence of IOP on axial length and refractive errors.

Unfortunately, POAG is difficult to diagnose in highly myopic eyes as the structural configuration of the optic nerve acts as a confounder variable in diagnostic tests such as spectral domain optical coherence tomography (SD-OCT), visual fields and electroretinography (ERG) [19,20,21,22]. Commercially available OCT devices still do not accurately measure RNFL thickness using automatic segmentation protocols in highly myopic eyes, exhibiting thinner average distributions in the superior, nasal, and inferior sectors, with greater temporal thickness, and a temporal shift in the superior and inferior peak locations [9].

To overcome the difficulties of structural diagnosis, OCT-based ganglion cell layer (GCL) analysis was proposed as a parameter for earlier detection of glaucomatous damage in myopic patients [23]. However, glaucoma is an aging-linked neurodegenerative pathology, so age could be a confounding factor in SD-OCT measurements. Indeed, animal studies showed contradictory results, with no definitive correlation between aging and thinning of the GCL [24,25,26,27] as detected in other neurodegenerative diseases [28], especially with the RNFL parameter [29,30,31]. Electrophysiological tests also showed functional impairment in patients affected by ocular hypertension, glaucoma, demyelinating optic neuropathies and Alzheimer’s disease in both human [32] and animal studies [33,34], with a functional electrical deficit followed by structural tomographic thinning [35], even without visual dysfunction [34]. Hence, there is ample scope for studying the physiopathology of neurodegenerative diseases, aging, myopia and glaucoma.

The aim of this study is to analyze the impact of chronic ocular hypertension (OHT) on the refractive error, structure and function of the neuroretina in two different OHT-inducing animal models, comparing it with a control cohort presenting ocular normotension (ONT) over 24 weeks.

## 2. Materials and Methods

A retrospective study was conducted by collecting data from a proprietary database of animal glaucoma projects (PI17/01946, MAT2017-83858-C2-2). To investigate the impact of IOP on the eye, 260 rat eyes were classified as inclusion criteria into ONT (if IOP was <20 mmHg) or OHT (if IOP was >20 mmHg) cohorts.

### 2.1. Animals

Long–Evans rats (40% male, 60% female) were used for the study. All animals were four weeks old, their weights ranged from 50–100 g at the start of the study and were similar to that reported by the supplier (Janvier-labs, Le Genest-Saint-Isle, France) over the study. The animal study was carried out in the experimental surgery department of the Biomedical Research Center of Aragon (CIBA). The experiments were previously approved by the Ethics Committee for Animal Research (PI34/17) and were carried out in strict accordance with the Association for Research in Vision and Ophthalmology’s Statement for the use of Animals. 

The control ONT cohort included non-intervened eyes. The OHT cohort comprised the episcleral sclerosis (ES) group, in which OHT was induced by biweekly sclerosis of the episcleral veins with hypertonic (1.8M) solution as described [36]. The microspheres (MS) group, in which OHT was induced by injecting poly-lactic-acid-glycolic (PLGA) microspheres into the anterior chamber at baseline biweekly for the first month and then once monthly until week 20 as described [35,37]. All OHT injections were performed in the right eye under surgical conditions: controlled temperature, topical tetracaine (1 mg/mL + oxibuprocaine 4 mg/mL) eye drops (Anestesico doble Colirccusi^®^, Alcon Cusí^®^ SA, Barcelona, Spain) and intraperitoneal (60 mg/kg of Ketamine + 0.25 mg/kg of Dexmedetomidine) anesthetic. Afterwards, the rats were left to recover in an enriched 2.5% oxygen atmosphere and were treated with antibiotic ointment (erythromycin 5 mg/g (Oftalmolosa Cusí^®^ eritromicina, Alcon Cusí^®^ SA, Barcelona, Spain)).

For detailed methodology characteristics, consult the original articles [27,35,37,38].

### 2.2. Intraocular Pressure

IOP measurements using a Tonolab^®^ tonometer (Tonolab, TiolatOy Helsinki, Helsinki, Finland) were recorded in the morning every week. The IOP value was obtained from the average of 18 rebounds. This procedure was accomplished using a sedative mixture of 3% sevoflurane gas and 1.5% oxygen for less than 3 min in order to avoid hypotension effects [39].

### 2.3. Optical Coherence Tomography

SD-OCT (Spectralis, Heidelberg^®^ Engineering, Heidelberg, Germany) was used to analyze the refractive status and the structure of the neuroretina over six months. Recordings were performed at the initial (0 weeks), middle (12 weeks) and end time of the study (24 weeks). In addition, an intermediate test was performed at week 8 of the study, which corresponds to 12 weeks of age of the rat. At this age, development and growth of the retina end and the retina reaches maturity [40]. A plane power polymethylmethacrylate (PMMA) contact lens with a thickness of 270 μm and a diameter of 5.2 mm (Cantor+Nissel^®^, Northamptonshire, UK) was adapted to the cornea to obtain high-quality images [41].

**Refractive status** was measured in diopters (D). The RNFL protocol was used for imaging acquisition. This protocol explores the optic nerve head, which is the most posterior structure of the rat eye, as the eye is elongated. Retinal images were adjusted and acquired by focusing on the retinal vascular structure. The diopters obtained through focusing were then analyzed as the diopter power of the eyeball. 

**Structural analysis:** The RNFL, retina posterior pole (R) and GCL protocol with automatic segmentation were used to quantify neuroretinal thickness (in micrometers). These protocols analyze an area of 1, 2 and 3 mm around the center of the optic disc using 61 scans. Subsequent follow-up examinations were acquired at the same location using the eye-tracking software and follow-up application. Biased examinations were discarded or manually corrected by a masked, trained technician if the algorithm had obviously erred. The R and GCL analyzed the 9 ETDRS areas (central area and inner or outer sectors divided into inferior, superior, nasal, and temporal sectors) and the total volume. The RNFL protocol analyzed 6 peripapillary sectors (inferotemporal, temporal, superotemporal, superonasal, nasal, and inferonasal).

For ERG and OCT acquisition, the rats were anesthetized by intraperitoneal administration of 60 mg/kg of Ketamine + 0.25 mg/kg of Dexmedetomidine.

### 2.4. Electroretinography

Electroretinography performed to analyze the outer and inner neuroretinal cells’ functionality at weeks 0, 12 and 24. The electroretinography device (Roland consult^®^ RETIanimal ERG, Brandenburg an der Havel, Germany) was used to stimulate simultaneously both eyes and explore them in multisteps using the flash scotopic and the photopic negative response (PhNR) protocols. Scotopic test examined rod response: step 1: −40 dB, 0.0003 cds/m^2^, 0.2 Hz (20 recordings averaged); step 2: −30 dB, 0.003 cds/m^2^, 0.125 Hz (18 recordings averaged); step 3: −20 dB, 0.03 cds/m^2^, 8.929 Hz (14 recordings averaged); step 4: −20 dB, 0.03 cds/m^2^, 0.111 Hz (15 recordings averaged). Photopic test examined retinal ganglion cell functionality: blue background 470 nm, 25 cds/m^2^ and red LED flash 625 nm, 0.30 cds/m^2^, 1.199 Hz (20 recordings averaged). For this purpose, the animals were prepared and the ERG tests were performed as [27,35] described. 

### 2.5. Statistical Analysis

Statistical analysis was performed by a blinded researcher. Descriptive analysis of quantitative variables was performed using (mean ± standard deviation [SD]). As the Kolmogorov–Smirnov test showed a no normal distribution of the variables, comparisons between both the ONT and OHT cohorts and ES and MS groups were conducted using the non-parametric Mann–Whitney U test and comparisons among the ONT, ES and MS cohorts were made using the Kruskal–Wallis H test. Eyes in every cohort were divided into two groups: those with a diopter power higher or lower than 17 D for the ONT eyes and 6 D for the OHT eyes. These sub-groups were used to study the logistic regression to the model so that correlations between refraction and OCT parameters or IOP values could be made. Correlations with the refractive status of the eye, age, OCT thicknesses, and ERG parameters were performed using Spearman correlation coefficient and logistic regression analysis. *p* values < 0.05 were considered statistical significative; the Bonferroni correction for multiple comparisons was also calculated to avoid a high false-positive rate.

## 3. Results

A total of 260 eyes of Long–Evans rats were analyzed from two different cohorts: 74 eyes with an IOP lower than 20 mmHg (ONT cohort serving as control), and 186 eyes with an IOP higher than 20 mmHg, named the OHT cohort, which in turn was divided into two sub-groups by induced OHT model: 62 eyes based on the ES model and 124 eyes based on the biodegradable MS model.

### 3.1. IOP Analysis

**Comparing ONT and OHT cohorts.** ONT eyes (16.09 ± 2.25 mmHg (range: 13.37–17.62 mmHg)) showed statistically significant (*p* < 0.05) lower IOP than OHT eyes (23.66 ± 1.40 mmHg (range 21.44–26.74 mmHg)). **Comparing both OHT groups**. The ES group presented the highest IOP values up to week 12, after which the trend reversed (*p* < 0.001) (Figure 1a). Analyzing by sex, males suffered from slightly higher IOP values in both the ONT and OHT groups, and reached statistical significance at weeks 8, 10 and 16 (*p* = 0.022, *p* < 0.001 and *p* = 0.023, respectively) (Figure 1b).

### 3.2. Refraction

**Comparing ONT and OHT cohorts**. Both the ONT and OHT cohorts experienced a physiological trend toward emmetropia throughout the follow-up, especially over the first weeks of the study; from 35.16 ± 6.38 D at baseline (Appendix A Appendix A) to 11.87 ± 3.21 D in the ONT and 11.98 ± 5.21 in the OHT at 8 weeks. However, the OHT cohort exhibited lower refractive power that reached statistical significance at week 24 (+4.54 ± 1.5 D in the ONT cohort vs.+0.88 ± 2.36 D in the OHT cohort, *p* < 0.001) (Figure 2). **Comparing both OHT groups**. A small difference between both OHT groups was also detected (*p* = 0.001), with slightly lower dioptric values for the ES group (+1.66 ± 1.53 D in MS vs. +0.10 ± 2.91 D in ES) (Figure 2a). No statistically significant differences were found in refraction between sexes (*p* > 0.05), although males consistently presented lower dioptric power throughout the follow-up, especially in the OHT cohort (Figure 2b).

### 3.3. Analysis of the Correlation between Refraction, Ocular Hypertension, and OCT Parameters

There were no statistically significant correlations between the OCT values and the refractive status of the eye. IOP > 20 mmHg at week 24 was positively correlated to lower dioptric values with a B coefficient of 4.01 (*p* = 0.01) for the OHT cohort vs. 4.65 (*p* = 0.009) for the ONT cohort. 

### 3.4. OCT Analysis

**Comparing ONT and OHT cohorts**. The most significant differences were found in week 8, when the OHT cohort exhibited thicker R values in the inner temporal sector and all the outer sectors (Table 1). The highest percentage differences by sector also followed the glaucomatous rule inferior > superior > nasal > temporal until week 12. The ONT cohort experienced a progressive decrease in RNFL thickness over time, and the OHT cohort showed an increase in RNFL thickness at week 12 in both the ES and MS groups (Figure 3). **Comparing both OHT groups**. Differences between the ONT and OHT cohorts were usually related to the values found in the MS group.

### 3.5. ERG Analysis

**Comparing ONT and OHT cohorts.** The ONT cohort presented wider variability in subject responses compared to the OHT cohort (Figure 4, Figure 5 and Figure 6). In the OHT cohort, a stronger response was found in scotopic and photopic conditions in week 12, with shorter latencies (Figure 4a, Figure 5a and Figure 6a) and greater amplitudes (Figure 4b, Figure 5b and Figure 6b). However, maintenance of hypertension over time (week 24) decreased the electrical signal in the ERG, which almost matched the ONT values, especially in the later stages. **Comparing both OHT groups**. Generally, the MS groups exhibited faster latency and greater amplitude in week 12 in scotopic and photopic conditions, although this trend reversed in week 24. 

## 4. Discussion

The effects of ocular hypertension have been described in a multitude of short-term animal studies, generally up to a maximum of 8 weeks of follow-up [37,42,43,44]. This study analyzes the effect of chronic OHT maintained over time focusing the study between 8 and 24 weeks of chronicity. To our knowledge, this is the first retrospective longitudinal 24-week study carried out in young-adult Long-Evans rats in vivo using automatically segmented OCT images that analyze the effect of mild-moderate OHT (between 20–30 mmHg) on refraction from the neuroretinal perspective (structure and function) and employs two different OHT-inducing models (ES vs. MS), in comparison to healthy controls. This allows us to analyze the neurodegenerative process in conditions of chronic OHT on myopia.

### 4.1. Refractive Analysis

The total optic power of the eye depends on factors such as axial length, corneal power and lens power. At birth, human eyes are usually hyperopic, and afterwards there is a trend toward emmetropia as the cornea structure stabilizes, though axial length may continue to grow until the second decade. However, in high myopia there is continuous, progressive axial elongation throughout life, possibly due to genetic, environmental and behavioral factors [45,46]. Although all the mechanisms involved in this progressive enlargement of the eyeball are not completely understood, the effect of IOP on axial elongation of the posterior pole [47,48], the importance of sclera stiffness [49] and the dynamic responses of the sclera after a chronic increase in IOP [50] have been factors involved in the emmetropization process. 

Ocular hypertension in congenital glaucoma occurs in developing loose tissue, meaning the sclera still exhibit great plasticity and stretch, increasing axial length [51]. This is similar to what occurs in high myopia [52,53], which supports the incidence of high myopia in adulthood [54]. When ocular hypertension occurs in older ages, scleral tissues are more rigid and stiffen [55], explaining the greater glaucomatous damage (IOP spikes are more poorly mitigated by less elastic sclera), the outward shift of the lamina cribosa and remodeling of connective tissue [56], and the deeper anterior chamber described in both the ES model and DBA/2J glaucomatous mice (not observed in OHT models using cauterization, probably due to the experimental intervention itself) [57,58]. Indeed, there is increasing evidence about the development of delayed and sustained OHT associated with repeated intravitreal anti-vascular endothelial growth factor injections in retinal diseases, such as age-related macular degeneration [59,60]. Several structure optic nerve head manifestations after intravitreal volume injections have been lately described as a mechanical displacement of the optic nerve head and canal expansion, with a widening and deepening of the optic, a prelaminar tissue thinning and a Bruch membrane opening expansion, suggesting structural changes after IOP and volume increase in aged patients, with rigid sclera [61]. Our experiments started when the rats were 4 weeks old, so they should not be considered either adults or newborns, but young rats. The refractive status was likewise not consistent with high myopia (−6 D) so, in our opinion, eye elongation was presumably more similar to what occurs in human POAG. Our OHT models supported the pre-existing evidence, as the ES cohort presented higher IOP, greater axonal damage, and lower dioptric power.

This study demonstrated the influence of sustained mild–moderate IOP on the emmetropization process. The baseline optical power was 35.16 ± 6.28 D (as [62] reported), with a progressive trend toward emmetropia in both ONT and OHT eyes. However, the statistically significant correlations to refraction were found only with mild–moderate OHT at week 24 in both ES and MS models, and were not found with functional or structural alterations. Therefore, a chronic sustained increase in IOP was considered the main risk factor involved in the loss of refractive power in OHT eyes. In this regard, animal and human studies also correlated hypotensive treatments such as trabeculectomy [63] and topical drugs such as latanoprost [16] to a decrease in total axial length, although studies with drugs such as timolol showed inconsistencies in both human [18] and animal models [17]. One of the next steps worth studying in future prospective studies is the influence of prompt normalization of IOP on refraction in these OHT models as a possible treatment for myopia. 

Nowadays, high myopia is increasingly prevalent in populations that carry out near-vision activities [64,65]. Sharp focus on a near image requires concomitant activation of both inferior and medial rectus muscles, and the subsequent compression of the eyeball by both muscles may cause an increase in IOP [66], similar to what happens in other compressive pathologies [67,68]. In addition, greatest drainage of aqueous humor occurs mainly via the nasal and lower episcleral veins [69,70,71,72,73], so sustained compression of these muscles at these locations during near-vision activities for extended periods of time would increase IOP. Moreover, in childhood and early adulthood, when eye elongation is easily achieved as it is less rigid, refractive power may consequently decrease [74]. Indeed, lower dioptric power would require higher demand for more synkinesis accommodation/convergence, creating a vicious circle [75]. Current studies with atropine drops are showing promising results in controlling myopia by producing a reversible anti-accommodative effect [76,77], and the dose-dependent anti-muscarinic effect on the smooth muscles of the episcleral veins could also balance the outflow.

In this study, the ES model, in which the outflow of the episcleral veins is limited, induced higher IOP and higher negative dioptric power. MS model produced a progressive alteration of the trabecular meshwork due to the mechanical clogging [78], but in a slower and more progressive structural and functional damage, more similar to that of the human POAG, and a lower negative dioptric power [35]. Moreover, a trend toward higher dioptric power was found in females, as well as lower levels of IOP, attributed to estrogens [79]. Scleral stiffness depends on choroidal vascular factors, among others, so estrogens could contribute to vascular protection against peripapillary rigidity and posterior location of the optic nerve head, which could explain the higher myopia rates found in male rats.

To our knowledge, this is the first study that longitudinally monitors eye growth in vivo (by indirect study of optical power) using non-invasive, simple, objective OCT technology that demonstrates progressive myopization in glaucomatous rats, influenced by sex and the OHT-inducing model. Refractive study in small animals usually requires invasive, expensive or low-reliability techniques such as enucleation, magnetic resonance imaging or ultrasound biomicroscopy, respectively, which makes longitudinal studies with the same animal impracticable. Previous ocular biometric studies using OCT on healthy [80,81] and glaucomatous mice [50,82] have been performed. However, only one study of healthy rats measured biometrics in vivo using high-resolution A-scan ultrasonography [62], and no convincing evidence of emmetropization during normal eye development was detected. Similar results were found in this study, in which the ONT cohort never experienced negative optical power (myopia), unlike rats in the OHT cohort. 

Analysis of refractive status constitutes a methodological limitation. As it was a retrospective study, it was not designed to analyze axial length, so the refractive status of the eye was measured with the OCT focuser, which improves image quality when acquiring OCT scans. The value obtained via this method is acceptable, but less accurate than axial length. Moreover, the use of a rigid contact lens for OCT imaging from which the retinal focusing data were obtained, introduces a tear lens, whose power will vary according to the mismatch between the fixed, posterior radius of curvature of the lens and anterior corneal radius, which can be expected to increase with age, and may also be altered in response to intracameral injections of MS. However, the lens was used in each examination, so the induced error would be constant, and the refractive trend found in our results would be constant throughout the study.

### 4.2. Structural Analysis

A decrease in RNFL thickness is a common feature in both myopic [83] and glaucomatous patients, though RNFL segmentation errors in automated SD-OCT analysis are frequent in myopia [84]. The good correlation observed between GCL and RNFL analysis has surpassed total retinal quantification, and GCL analysis has emerged as a more appropriate imaging tool for detecting early glaucomatous progression in myopic patients [85].

When analyzing the R, RNFL and GCL values, OCT protocols were performed around the optic disc as rats do not have macula, so central values could lead to misinterpretations. This is one of the main reasons why correlations to human retina are limited. As regards general trends, differences in R thickness between the ONT and OHT cohorts were seen up to week 8, although they then decreased through week 24, possibly due to a neurodegenerative process, with an initial inflammatory response that causes an increase in thickness (activated microglia and other inflammatory mediators [31]), and a final atrophic pattern. The greater thickness found in external sectors of peripheral retina could be attributed to peripheral immune infiltration [31]. On the other hand, it is important to highlight that there were no correlations between refraction and R, GCL, and RNFL thicknesses. IOP was therefore the most important factor modifying retinal thickness, meaning that the RNFL thinning observed in the OHT cohorts in week 24 was secondary to neurodegeneration rather than a stretching of the retinal tissue. Hence, OCT has demonstrated its reliability when analyzing retinal structures in neurodegenerative diseases [28], even during the emmetropization process.

Comparing both OHT groups, the ES group exhibited thicker R, RNFL and GCL values at week 8, but the thickness loss rate was also higher over time, reaching the lowest thickness at the end of follow-up. As this cohort suffered from higher IOP levels, these eyes could have suffered intense damage (a more extreme response) at GCL level that explains this drop. Conversely, RNFL analysis of the OHT cohorts showed an initial increase in thickness until week 12, after which the trend reversed in both groups, although it did so more dramatically in the ES group, supporting this theory.

The lack of histological studies is the main handicap of structural analysis, but several studies reflected the good correlation between immunohistochemistry and OCT thicknesses, making SD-OCT reliable for research in retinal degeneration [28]. Another limitation of this study was not having considered the lateral magnification that occurs as the anterior chamber increases, which overstates the retinal thickness. Most myopic eyes may therefore have even smaller retinal thicknesses [86].

### 4.3. Functional Analysis

ERG showed supranormal responses in the OHT cohorts, especially in the MS group in week 12. These outcomes were also observed in other animal studies with acute [43,47] and chronic [48] non-ischemic increases in IOP. This noxa could lead to initial hyperstimulation of the neuronal structures, especially bipolar and intermediate cells, while the process is neither ischemic nor chronic. In other words, we are possibly witnessing the beginning of the disease in an early or even reversible phase. However, this initially stronger electrical response disappeared at week 24, suggesting the decline of this phenomenon, probably due to ischemia or neurodegenerative damage, as occurs in POAG. It has also been hypothesized that the supranormal ERG responses could be related to the increase in illuminated retinal area during the test (greater axial length means greater retinal area, compared to normal axial length), though this hypothesis could not explain all the ERG findings [44], among them the fact that the supranormal electrical response is not sustained over time, but only when the eye is more hyperopic (in early phases of the study). In contrast, human studies showed a negative correlation between axial length (but not refractive error) and the values of ERG responses in healthy adults [20,87], as occurs here at later stages. This pattern of electrical response was more extreme in the MS group and was consistent throughout the steps, showing a coherent pattern of electrical behavior in photoreceptor and bipolar cells. However, because the ES model is more aggressive, this suprasignal could have occurred before week 12 in the ES model. We may therefore have witnessed an ongoing decay phase.

This functional behavior could be somehow correlated to the structural OCT findings of this study. At week 12, the GCL and RNFL are not as damaged, and their status is reversible, which could be correlated to the ERG hyperresponsive pattern. This could be supported by a previous study (also using Long–Evans rats) with a similar electrical pattern after chronic IOP elevation (by circumlimbal suture) that returned to baseline after suture removal [88]. This functional damage was not correlated with structural damage to GCL density, but with a reduction in the RNFL in week 15 in the post-hoc analysis [88]. Again, it is important to note that the only parameter that was statistically correlated to the refraction was IOP > 20 mmHg, not the ERG parameters. Our results therefore suggest that refractive status also appears to be an independent factor for ERG tests when analyzing POAG.

## 5. Conclusions

In this article, we pointed out the relationship between ocular hypertension and refraction over time. Visual disturbances in glaucomatous patients could also be associated with a progressive negative refractive error, in addition to the neurodegenerative damage associated with glaucoma, so frequent assessment of refraction may be important in these patients over time. The study of functional and structural tests in early and late stages in two different POAG models produced a deeper understanding of the pathophysiological retinal damage caused by the increase in IOP, with an initial increase in electrical signal associated with a thickening pattern detected using OCT, followed by functional and structural impairment. In addition, the lack of correlation between the refractive status of the eye and the structure and function of the retina provided further insight into the usefulness of OCT and ERG in relation to the emmetropia process.

## Figures and Tables

**Figure 1 jcm-10-03697-f001:**
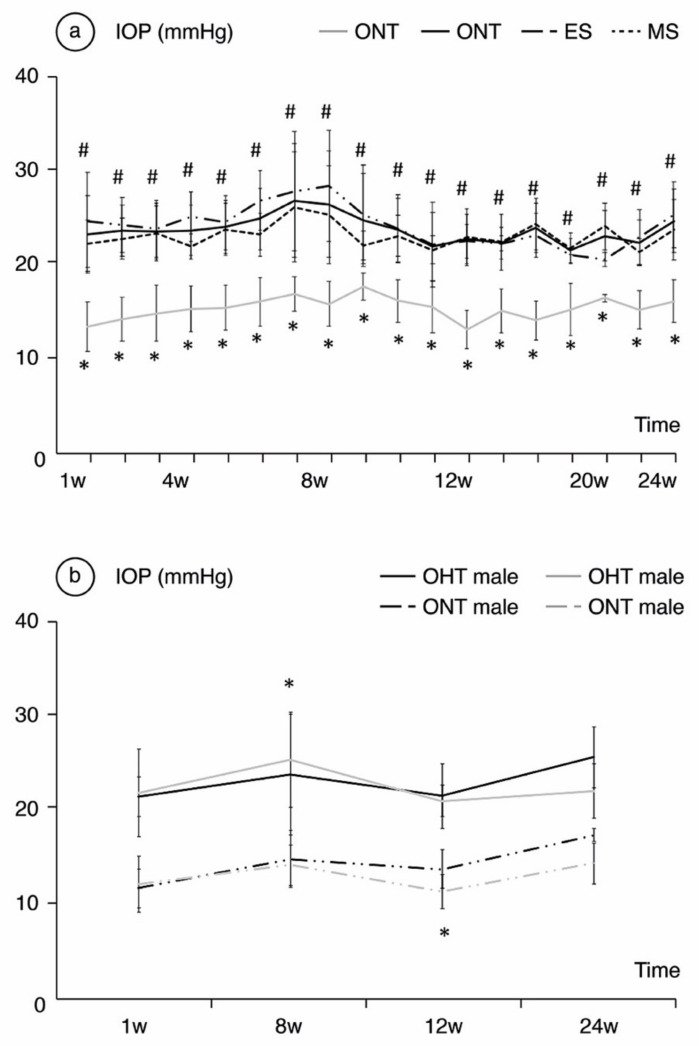
Analysis of intraocular pressure (IOP) in mmHg over the 24-week follow-up comparing the ocular normotension (ONT) and ocular hypertension (OHT) cohorts (**a**) and comparing both sexes (**b**). Abbreviations: w: week; ES: episcleral sclerosis group; MS: microspheres group; * statistical differences between the ONT and the OHT cohorts; #: statistical differences between the ES and MS groups.

**Figure 2 jcm-10-03697-f002:**
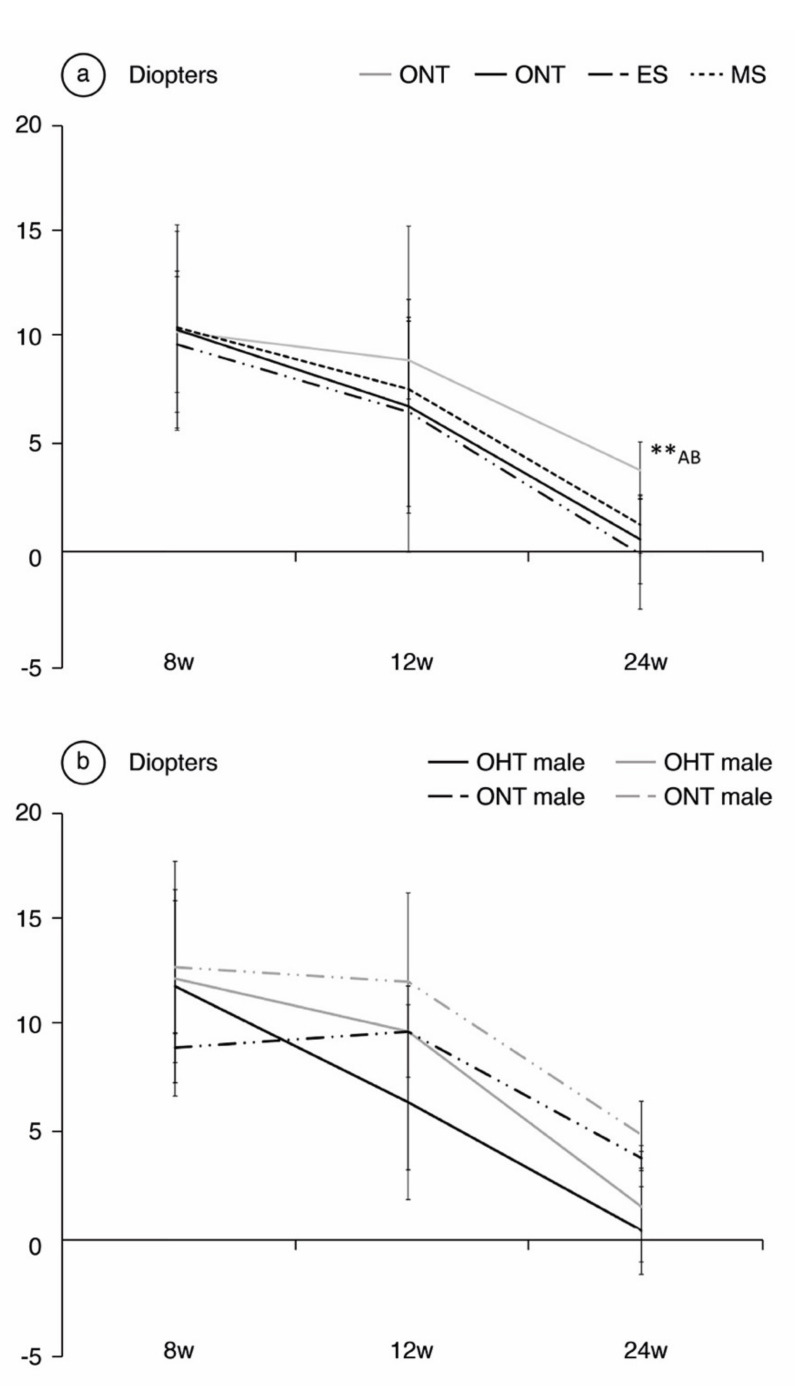
Analysis of refractive power in diopters over the 24-week follow-up, comparing the ocular normotension (ONT) and ocular hypertension (OHT) cohorts (**a**) and comparing both sexes (**b**). Abbreviations: w: week; ES: episcleral sclerosis group; MS: microspheres group; **: statistical differences between groups (ONT vs.ES vs. MS) (Kruskal Wallis); A: statistical differences between ONT and ES; B: statistical differences between ONT and MS.

**Figure 3 jcm-10-03697-f003:**
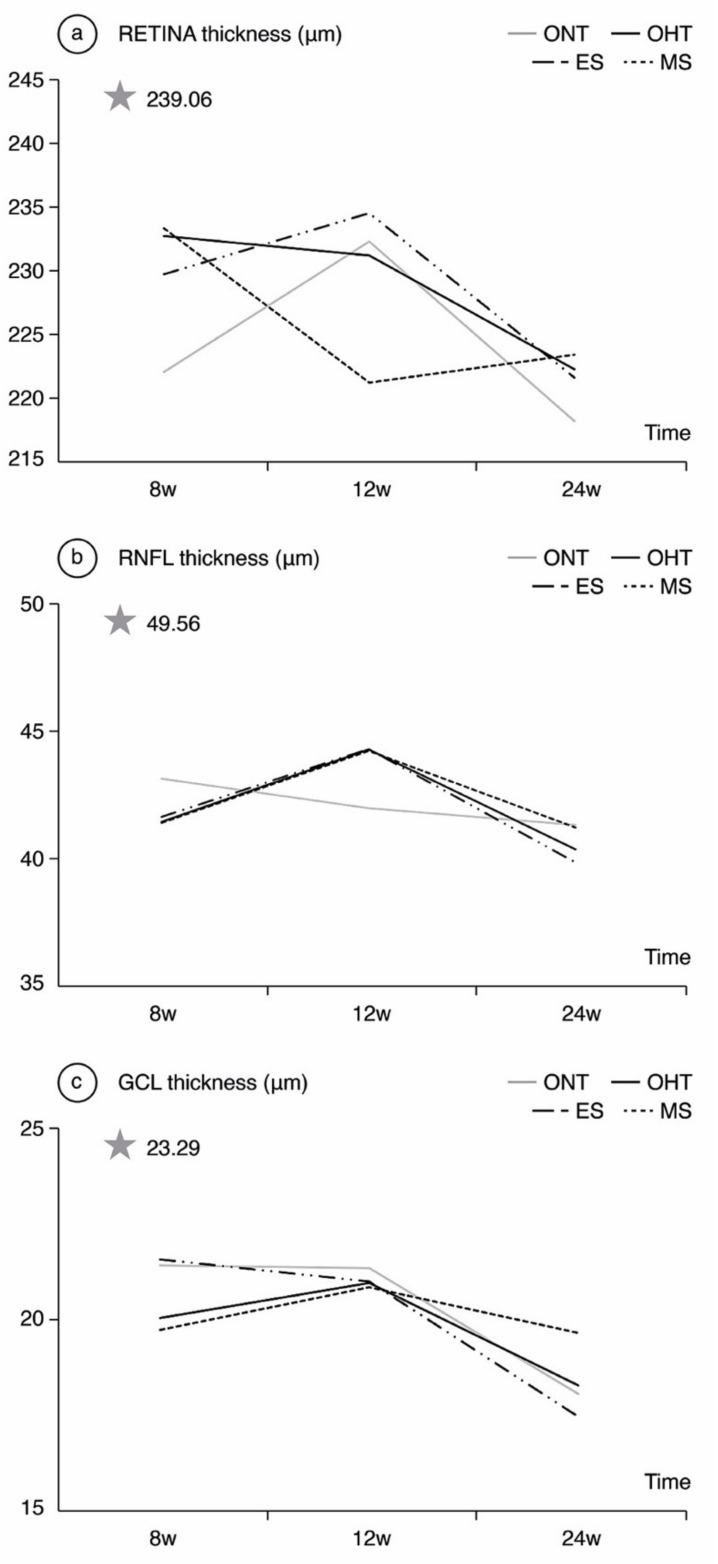
Neuroretinal thickness measured by optical coherence tomography in both the ocular normotension and ocular hypertension cohorts and the ocular hypertension sub-groups at 8, 12, 24 weeks follow-up. (**a**) Retina thickness, (**b**) Retina nerve fiber layer thickness, (**c**) Ganglion cell layer thickness. The analysis is based on the mean values of all the OCT sectors. ONT: ocular normotension cohort; OHT: ocular hypertension cohort; ES: episcleral sclerosis group; MS: microspheres group; RNFL: retinal nerve fiber layer; GCL: ganglion cell layer; μm: thickness in micrometers; W: week; star: baseline thickness.

**Figure 4 jcm-10-03697-f004:**
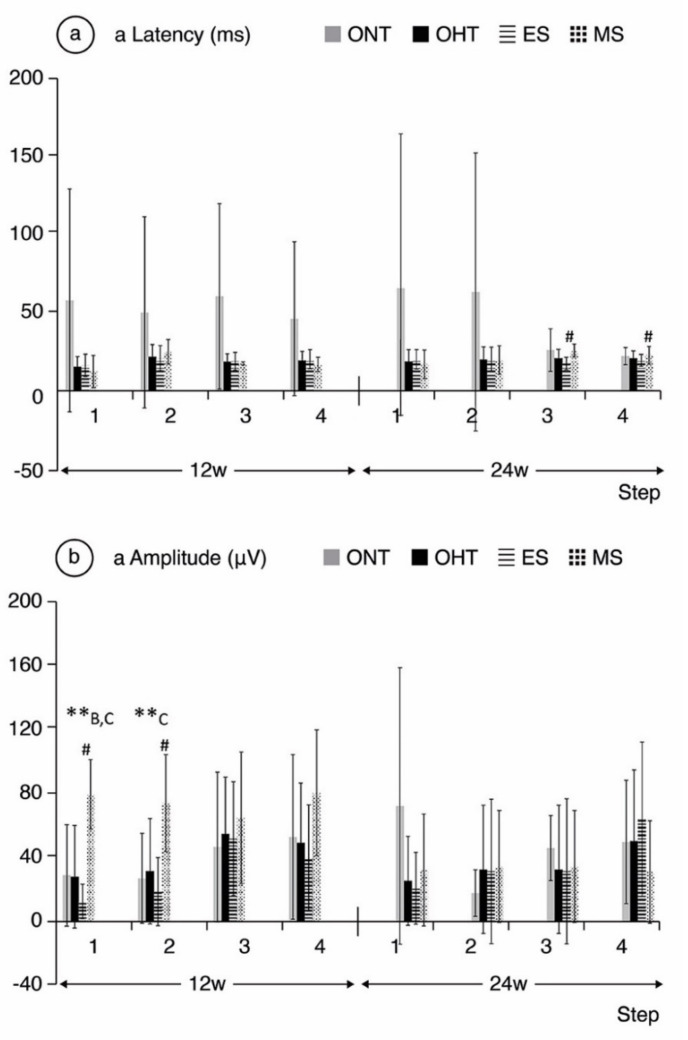
a-wave electroretinographic response in both the ocular normotension and ocular hypertension cohorts and the ocular hypertension sub-groups at weeks 12 and 24. (**a**) a-latency in milliseconds of photoreceptors under scotopic conditions. (**b**) a-amplitude in microvolts of photoreceptors under scotopic conditions. Abbreviations: ONT: ocular normotension group; OHT: ocular hypertension group; ES: episcleral sclerosis group; MS: microspheres group; w: week; ms: milliseconds; µV: microvolts; **: statistical differences between groups (ONT vs. ES vs. MS) (Kruskal–Wallis); #: statistical differences between the ES and MS groups; A: statistical differences between ONT and ES; B: statistical differences between ONT and MS; C: statistical differences between ES and MS. Scotopic test (rod response): step 1: −40 dB, 0.0003 cds/m^2^, 0.2 Hz (20 recordings averaged); step 2: −30 dB, 0.003 cds/m^2^, 0.125 Hz (18 recordings averaged); step 3: −20 dB, 0.03 cds/m^2^, 8.929 Hz (14 recordings averaged); step 4: −20 dB, 0.03 cds/m^2^, 0.111 Hz (15 recordings averaged).

**Figure 5 jcm-10-03697-f005:**
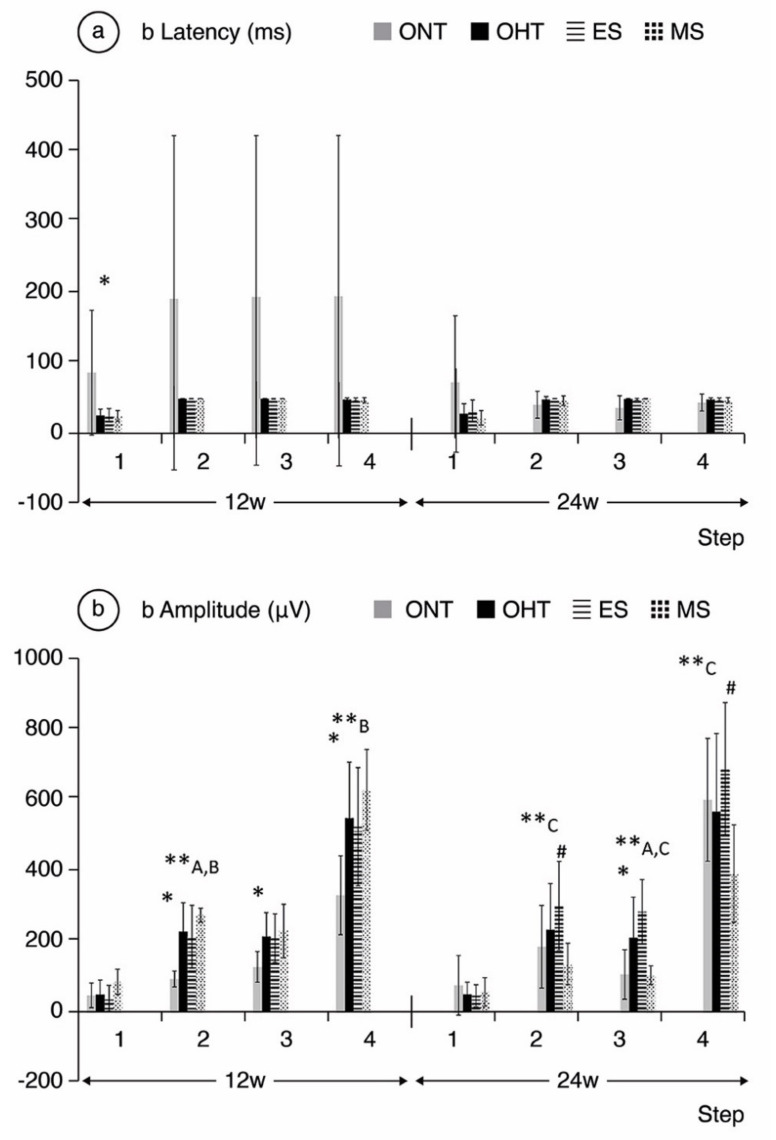
b-wave electroretinographic response in both the ocular normotension and ocular hypertension cohorts and the ocular hypertension sub-groups at weeks 12 and 24. (**a**) b-latency in milliseconds of intermediate cells under scotopic conditions. (**b**) b-amplitude in microvolts of intermediate cells under scotopic conditions. ONT: ocular normotension group; OHT: ocular hypertension group; ES: episcleral sclerosis group; MS: microspheres group; w: week; ms: milliseconds; µV: microvolts; * statistical differences between the ONT and OHT cohorts. **: statistical differences between groups (ONT vs. ES vs. MS) (Kruskal–Wallis); #: statistical differences between the ES and MS groups; A: statistical differences between ONT and ES; B: statistical differences between ONT and MS; step 1: −40 dB, 0.0003 cds/m^2^, 0.2 Hz (20 recordings averaged); step 2: −30 dB, 0.003 cds/m^2^, 0.125 Hz (18 recordings averaged); step 3: −20 dB, 0.03 cds/m^2^, 8.929 Hz (14 recordings averaged); step 4: −20 dB, 0.03 cds/m^2^, 0.111 Hz (15 recordings averaged).

**Figure 6 jcm-10-03697-f006:**
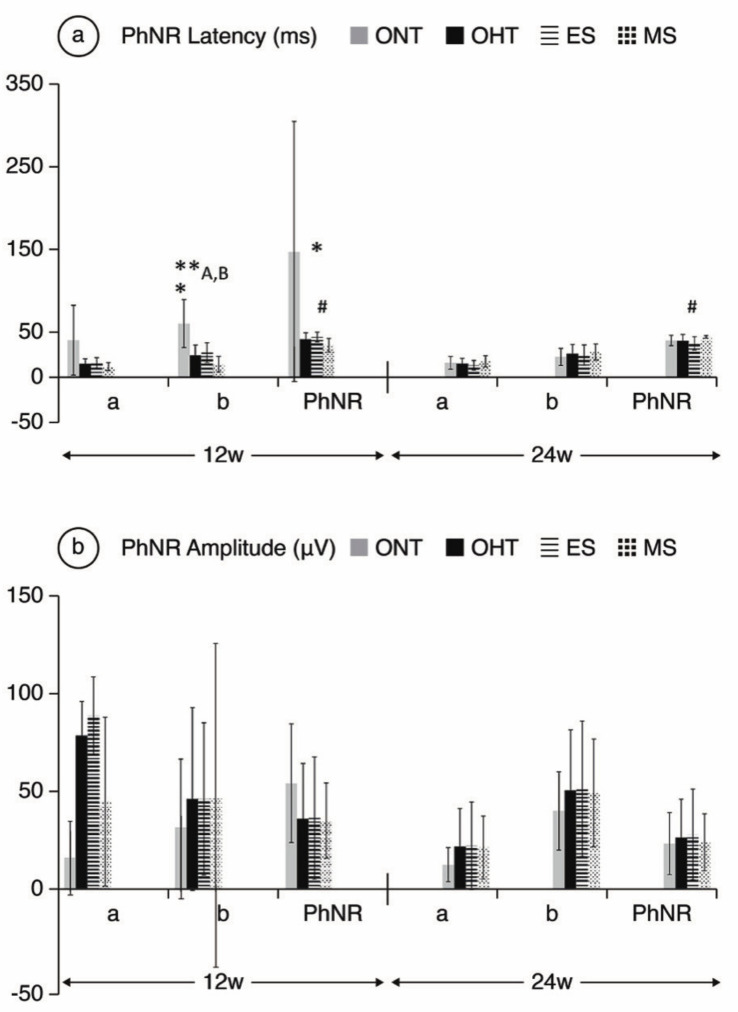
PhNR wave electroretinographic response in both the ocular normotension and ocular hypertension cohorts and the ocular hypertension sub-groups at weeks 12 and 24. (**a**) PhNR latency of in milliseconds under photopic conditions. (**b**) PhRN amplitude in microvolts under photopic conditions. Abbreviations: ONT: ocular normotension group; OHT: ocular hypertension group; ES: episcleral sclerosis group; MS: microspheres group; w: week; ms: milliseconds; µV: microvolts; * statistical differences between the ONT and OHT cohorts. **: statistical differences between groups (ONT vs. ES vs. MS) (Kruskal–Wallis); #: statistical differences between the ES and MS groups; A: Statistical differences between ONT and ES; B: Statistical differences between ONT and MS. Photopic test: blue background 470 nm, 25 cds/m^2^ and red LED flash 625 nm, 0.30 cds/m^2^, 1.199 Hz (20 recordings averaged). a: photoreceptors response; b: intermediate cells response; PhNR: ganglion cells response.

**Table 1 jcm-10-03697-t001:** Statistically significant differences found by optical coherence tomography parameters analyzed at weeks 8, 12 and 24.

Time	OCT Protocol	Sector	ONT (Mean ± SD in μm)	OHT (Mean ± SD in μm)	D%	*p* *		OHT Groups (Mean ± SD in μm)	*p* ^†^
**8 W**	R	Central	277.20 ± 17.86	**263.21 ± 22.03**	−5.05	0.022	MS	263.80 ± 22.89	0.022
							ES	**260.30 ± 17.86**	
		Temporal Inner	**244.70 ± 8.35**	251.90 ± 18.50	+2.86	0.047	MS	**250.78 ± 14.82**	0.047
							ES	257.50 ± 31.74	
		Inferior Outer	**239.90 ± 6.19**	250.50 ± 11.13	+4.23	0.003	MS	251.40 ± 11.34	0.003
							ES	**246.00 ± 9.30**	
		Nasal Outer	**243.30 ± 4.34**	253.60 ± 12.85	+4.06	0.002	MS	253.96 ± 11.84	0.002
							ES	**251.90 ± 17.60**	
		Superior Outer	**246.30 ± 4.19**	256.86 ± 17.28	+4.13	0.017	MS	257.04 ± 16.69	0.017
							ES	**256.00 ± 20.89**	
		Temporal Outer	**245.00 ± 6.48**	254.10 ± 16.10	+3.58	0.013	MS	**253.12 ± 12.17**	0.013
							ES	259.00 ± 29.37	
	GCL	Central	19.80 ± 3.39	**17.03 ± 3.34**	−13.99	0.014	MS	**16.78 ± 3.27**	0.014
							ES	18.30 ± 3.62	
		Superior Inner	24.10 ± 1.85	**20.91 ± 4.03**	−13.24	0.004	MS	**20.53 ± 4.04**	0.004
							ES	22.80 ± 3.61	
**12 W**	R	Central	285.33 ± 18.90	**266.00 ± 16.73**	−6.76	0.039	MS	267.33 ± 17.21	0.039
							ES	**265.56 ± 17.61**	
**24 W**	RNFL	Nasal Superior	39.27 ± 7.25	28.88 ± 11.99	−26.46	0.036	MS	31.00 ± 10.66	0.050
							ES	**27.60 ± 13.10**	

OCT: optical coherence tomography; R: retina; GCL: ganglion cell layer complex; RNFL: retinal nerve fiber layer; W: week; ONT: ocular normotension cohort; OHT: ocular hypertension cohort; ES: episcleral sclerosis group; MS: microspheres group; D%: differences in percentage; *p*: statistical differences (<0.05): * U Mann Whitney; ^†^ Kruskal Wallis; the lowest thickness values are highlighted in bold.

## Data Availability

The data are available only upon request to the corresponding author.

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
