# Peer review of "Influence of Chronic Ocular Hypertension on Emmetropia: Refractive, Structural and Functional Study in Two Rat Models"

_jcm, 2021, doi:10.3390/jcm10163697_

Round 1

Reviewer 1 Report

In this study, the authors want to evaluate the effect of mild-moderate chronic OHT on refraction and neuroretina (structure and function) in young-adult Long-Evans rats. Overall, the study is very well conceived, however, there are some points that did not allow me to follow the results.

Despite this is a retrospective study based on methodology done in references 27, 35-37, I think that is missing a general overview of the methods of the animal models. This is particularly important to understand the time-points selected by the authors to perform the OCT (weeks 8, 12, 24) and ERG (weeks 0, 12 and 24).

Moreover, in the panels of results is hard to get information in order to follow the description of the results. For sure I suggest to the authors that the graphs need to be increased (Figure 1,2 and 3). For instance, for figure 1 the results can be split into two different figures, one for IOP and another one for refractive powers in diopters. Figure 3 has a lot of small graphs, and each graph has a lot of information, which makes this figure hard to extract information. The graphs in this figure should be divided. Even the legend of figure 3 contains a big description… why did the authors include a description of the results in this legend?

Line 187: The reference to table 1 should be done within the text with the result description and not in the title of the section.

Author Response

Manuscript ID: jcm-1326332

Dear Editor

Please find attached the revised (changes marked) and the final version of the paper entitled “Influence of chronic ocular hypertension on emmetropia: refractive,
structural and functional study in two rat models.”

Thank you very much for the revision. We are pleased to be reviewed by experts within the area, and will try our best to respond to their suggestions adequately.

We have made a special effort in clarify all the points marked by the reviewers. So the article has been generally improved.

We sincerely hope that the requests for changes have been properly addressed and that the revised manuscript is now found acceptable for publication.

Best regards

Silvia Méndez-Martínez

Reviewer 2 Report

The manuscript is well written and very interesting. 

However, in my opinion, this passage should be removed 

Authors should discuss the results and how they can be interpreted from the per- 408 spective of previous studies and of the working hypotheses. The findings and their impli- 409 cations should be discussed in the broadest context possible. Future research directions 410 may also be highlighted.

Author Response

Manuscript ID: jcm-1326332

Dear Editor

Please find attached the revised (changes marked) and the final version of the paper entitled “Influence of chronic ocular hypertension on emmetropia: refractive, structural and functional study in two rat models.”

Thank you very much for the revision. We are pleased to be reviewed by experts within the area, and will try our best to respond to their suggestions adequately.

We sincerely hope that the requests for changes have been properly addressed and that the revised manuscript is now found acceptable for publication.

Best regards

Silvia Méndez-Martínez
